# Research on Apple Recognition Algorithm in Complex Orchard Environment Based on Deep Learning

**DOI:** 10.3390/s23125425

**Published:** 2023-06-08

**Authors:** Zhuoqun Zhao, Jiang Wang, Hui Zhao

**Affiliations:** 1School of Electrical and Information Engineering, Tianjin University, Tianjin 300072, China; zhuoqunzhao@tju.edu.cn (Z.Z.); jiangwang@tju.edu.cn (J.W.); 2School of Mechanical Engineering, Tianjin Sino-German University of Applied Sciences, Tianjin 300350, China; 3School of Electrical Engineering and Automation, Tianjin University of Technology, Tianjin 300384, China

**Keywords:** target detection, recognition algorithm, joint loss function, soft NMS algorithm, global and local characteristics

## Abstract

In the complex environment of orchards, in view of low fruit recognition accuracy, poor real-time and robustness of traditional recognition algorithms, this paper propose an improved fruit recognition algorithm based on deep learning. Firstly, the residual module was assembled with the cross stage parity network (CSP Net) to optimize recognition performance and reduce the computing burden of the network. Secondly, the spatial pyramid pool (SPP) module is integrated into the recognition network of the YOLOv5 to blend the local and global features of the fruit, thus improving the recall rate of the minimum fruit target. Meanwhile, the NMS algorithm was replaced by the Soft NMS algorithm to enhance the ability of identifying overlapped fruits. Finally, a joint loss function was constructed based on focal and CIoU loss to optimize the algorithm, and the recognition accuracy was significantly improved. The test results show that the MAP value of the improved model after dataset training reaches 96.3% in the test set, which is 3.8% higher than the original model. F1 value reaches 91.8%, which is 3.8% higher than the original model. The average detection speed under GPU reaches 27.8 frames/s, which is 5.6 frames/s higher than the original model. Compared with current advanced detection methods such as Faster RCNN and RetinaNet, among others, the test results show that this method has excellent detection accuracy, good robustness and real-time performance, and has important reference value for solving the problem of accurate recognition of fruit in complex environment.

## 1. Introduction

Robots can automatically pick fruits, and solve the problems of high cost, low efficiency, and labor shortage [1,2]. The efficiency of a robot is related to its speed, accuracy, and adaptability to complex environments [3,4]. The solution of the above problems depends on the adoption of advanced fruit recognition theories and algorithms. At present, most of the existing fruit recognition algorithms are based on the laboratory environment, neglecting to consider the impact of the actual orchard complex environment on the accuracy and speed of fruit recognition. Meanwhile, with the development of artificial intelligence theory and technology, the recognition theory and algorithm based on deep learning are constantly making new progress. Therefore, it is of great academic significance and practical application value to study the application of deep learning in the field of fruit recognition and improve the existing theories and algorithms according to the practical needs.

At present, important progress has been made in the research of fruit recognition algorithm. Lin et al. [5] studied an image segmentation algorithm by means of Bayesian classifier and density clustering. On this basis, they proposed an orange recognition method using support vector machine, which can weaken the influence of illumination variation on the recognition accuracy, the disadvantage is slow recognition speed and poor real-time performance. Song Huaibo [6] put forward a method of apple detection based on convex Hull theory; the recognition accuracy of apple in occlusion condition is improved, but the recognition effect is easily affected by changes in lighting and color saturation. The above recognition methods are too dependent on the features extracted by humans; their results are susceptible to the influence of the complex environment, the robustness and generalization are poor, and there is a certain gap with the actual work needs of picking robots [7,8]. As deep learning continues to make new progress, the advantages of convolution neural networks (CNN) in extracting multi-level features are increasingly evident; its powerful representation ability greatly improves the effect of target recognition, and promotes the development of target recognition technology [9,10]. Its application field covers: motion action recognition [11], remote sensing scene recognition [12], video moment retrieval with noisy labels [13], etc. Currently, the common target recognition algorithms based on deep learning are the region suggestion algorithm and the non-region suggestion algorithm. The most well-known algorithms of the region suggestion algorithm are RCNN [14] (Regions with CNN features), Fast RCNN [15] and Faster RCNN [16], etc. Its main idea is to obtain the suggestion region first, and then make further classification and location prediction in the current region, this method is called two-order target detection [17,18]. Sun et al. [19] put forward an novel tomato recognition algorithm using Faster RCNN, in which ResNet50 and K-means clustering are used for feature extraction and adjusting the pre-boxes, respectivel; as a result, the recognition effect is effectively improved, the disadvantage is the slow recognition speed. Zhang Lei [20] studied the fast RCNN algorithm to realize the recognition of many kinds of fruits in the natural environment. This algorithm has high precision and strong generalization ability; however, the detection efficiency of overlapping occluded fruits is low, and it is easy to determine the overlapping fruits as one. Non-region suggestion algorithm (SSD [21], Yolo [22], etc.) can directly give classification results and target position coordinates through a single convolution neural network, also known as first-order target detection. Tian et al. [23] improved the YOLOv5 algorithm and used Dense Net as its feature extraction network to accurately identify apples at different stages of growth. However, the current research on fruit recognition in a wide field of vision is still insufficient. Zhao Dean et al. [24] improved YOLOv5 algorithm based on solving apple recognition problem in a complex environment; its recognition speed reaches 60 frames/s, which has a good real-time performance, but when detecting densely distributed apples, the F1 value drops obviously; thus, the method does not test well for densely distributed and wide field apples.

The main contributions of this paper can be summarized as follows: (a) In the complex environment of orchards, in view of low fruit recognition accuracy, poor real-time and robustness of traditional recognition algorithms, this paper propose an improved fruit recognition algorithm based on deep learning. (b) Targeted improvements were made to the YOLOv5 network, and the accuracy and real-time performance of the improved recognition algorithm are verified by testing in a complex environment. (c) To further verify the effectiveness and feasibility of the improved algorithm, the performance comparison test of the algorithms were carried out under different fruit quantities and light conditions; the test results prove the superiority of the improved algorithm.

The rest of this paper is organized as follows: Data processing is presented in Section 2 and includes “Data collection process and image sources, production of dataset and enhanced processing of data”. Section 3 discusses an improved apple recognition algorithm. Section 4 describes the method and process of model training. Section 5 and Section 6 present the results of the experiment and the final conclusions of the study, respectively.

## 2. YOLOV5s Network

YOLOv5 algorithm [25] is an advanced target detection model, which is fast, accurate and lightweight. The YOLO algorithm uses a single convolutional neural network (CNN) to process the entire image, dividing it into grids and assigning each grid to detect and predict the objects in its area. The YOLOv5 algorithm comes in four different versions that vary in model size and accuracy, ranging from small (YOLOv5s) to large (YOLOv5x). In this paper, we use the YOLOv5s model as the benchmark model for training due to its balance of size and accuracy.

The YOLOv5s network consists of four main parts. The first part is the input layer, which processes the input images. This network uses 608 × 608 pixel input images, which are preprocessed through Mosaic data enhancement, adaptive anchor box calculation, and adaptive image scaling to improve the training speed and accuracy of the model. The second part is the backbone network which consists of a series of convolutional layers, mainly using the CSPParkNet53 network and the Focus network to extract common features in the image. The third part is the neck network, which is a combination of the feature pyramid network (FPN) and path aggregation network (PAN). The FPN layers transmit strong semantic features from the top layers to the bottom layers, while the PAN layers transmit strong positioning features from the bottom layers to the top layers. This allows for better transmission of positioning information from the bottom layers to the top layers, improving the performance of the detection network. The neck network is essential for achieving high accuracy in object detection, as it helps to combine the semantic and positional information extracted by the backbone and output layers. The final part is the predictive output layer, which is responsible for making predictions about the objects in the input images. The predictive output layer consists of two branches: a classification branch and a regression branch. The classification branch predicts the class of each object in the input image, while the regression branch predicts the bounding box coordinates for each object. The output layer uses the processed features from the neck network to make these predictions, which are then used to identify and locate the objects in the input image. Overall, the YOLOv5s network is a powerful and efficient tool for object detection tasks.

## 3. Data Processing

### 3.1. Data Collection Process and Image Sources

In total, 2200 frames of apple images with a resolution of 4032 × 3024 pixels were collected under sunny and cloudy weather during four time periods, that is, morning, noon, evening and night; illumination conditions include front, side, back and artificial lighting. The robot picking process was simulated by constantly changing the angle and distance of shooting. The collected images include different colors, posture, size, illumination, background, overlap and occlusion. Because the collected apple images are mainly red Fuji, and the orchard environment is relatively simple, some apple images of different varieties (Red Fuji, Gala, snake fruit, Qin Guan, etc.) in the complex environment of other orchards and different maturity periods are collected through the Internet to enhance the universality of the visual system, and 1800 images are selected. Finally, a total of 4000 apple images were obtained through field shooting and Internet collection, which were uniformly saved in a Jpg format. Figure 1 shows some of the collected images.

### 3.2. Production of Dataset

In order to identify apples in different mature periods, apples are divided into mature and immature types according to the different growing periods. The training model of this study adopts Pascal VOC dataset format; with the help of labeling software, the scope where the target apple is located in the image is manually marked as a rectangular frame, the category label of mature apple is set as apple, and the category label of immature apple is set as green apple. After labeling, 3200, 400 and 400 images were randomly selected to form the training set, verification set and test set, respectively.

### 3.3. Enhanced Processing of Data

Mosaic data enhancement algorithm [26] is introduced into the input of the model, that is, four images are read each time, and operations such as random scaling, flipping, cropping and optical transformation are carried out, respectively; then, four images are stitched together and transferred into the network with the adjusted tag. It is equivalent to passing in four images at a time for learning, which further enriched the background of the target, and increased the number of images. The calculation of four image data can be synchronized in the standardized BN; this amounts to scaling up batch processing, and makes the mean and variance obtained according to BN layer closer to the distribution of the whole dataset, making the mean and variance obtained according to BN layer closer to the distribution of the whole dataset. Therefore, the robustness of the model is improved effectively.

## 4. Improved Apple Recognition Algorithm

### 4.1. CSP_X Module [27]

Drawing on the residuals idea of Res Net, residual connections are heavily used in YOLOv5’s backbone network; the maximum pooling layer is replaced by a 3 × 3 convolution with stride 2 for down sampling. Therefore, the network can be designed deeply, thus being able to retain more characteristic information, and eliminate network degradation (such as gradient disappearance and explosion) in training. The above improvements make the model easier to converge and have stronger feature extraction capabilities. Its network structure is mainly composed of a series of residual blocks in series. The residual block structure is shown in Figure 2. In the picking process, real-time performance is an important indicator. Therefore, it is very necessary to reduce the amount of computation by making the network more lightweight.

Cspnet (Cross stage parallel network) is an advanced network which can eliminate gradient information repetition. The reuse of gradient information is avoided, and the floating points operations (FLOPs) is effectively reduced; in the case of ensuring the reasoning speed and accuracy, the computation of the model is also reduced. Therefore, the residual blocks are grouped with CSP Net to form the CSP_X module (see Figure 3). With the help of the CSP_X module, the input feature map is subsampled twice by the first CBL module and divided into two branches, the first of which is processed by the other CBL module; then, through X residual modules and convolution processing, another branch directly undergoes convolution processing; finally, the results of these two parts are spliced through channels to obtain the output feature map.

### 4.2. Spatial Pyramid Pool SPP Module [28]

Although Yolov5 combines the deep and shallow features and enhances the multi-scale performance of the model, the recognition accuracy for small objects is still insufficient, and the miss rate is higher. Therefore, in order to accurately identify the small apple target in a large field of view, it is necessary to pre-position the overall distribution of the apple, guide the robot to plan the path, adjust the posture, and complete the goal of counting and estimating the apple yield [29,30]. The SPP (spatial pyramid pool) module is integrated into YOLOv5, and it is a structure composed of 3 max-pooling layers with different scales, as shown in Figure 4. The SPP module firstly performs down-sampling operation on the input characteristic graph through three maximum pooling layers with different pooling kernel sizes. Then, its result and the original input feature map are spliced by channel. Finally, the characteristic graph is spliced and sent to the subsequent network, which can effectively fuse local features and global features, expand the receptive field, enhance the expressive ability of the characteristic graph, and further improve the recognition accuracy for micro target.

### 4.3. Soft NMS (Non-Maximum Suppression) Algorithm [31]

In the existing object detection algorithms, for the same real object, it is often necessary to output multiple prediction frames to ensure a recall rate. Because of the existence of redundant prediction frames, the recognition accuracy will be reduced; therefore, with the help of NMS algorithm, overlapping prediction frames are filtered to obtain the best prediction output [32]. For the traditional NMS method, if the prediction score of a frame is higher, this frame will be given priority, and other frames that overlap with it will be discarded if they exceed a certain threshold. Although this method is simple and effective, in the actual picking environment, when the fruits are denser, those with lower scores may be suppressed, resulting in missed detection. To solve the above problems, the traditional NMS is improved to a soft NMS algorithm, as follows:(1)$$S_{i}=\left\{\begin{array}{cc} S_{i}, & \;\mathrm{IoU}(M,b_{i})<N_{t}\\ S_{i}=S_{i}e^{-\frac{\mathrm{IoU}{\left(M,b_{i}\right)}^{2}}{\sigma }} & ,\;\;\mathrm{IoU}(M,b_{i})\geq N_{t} \end{array}\right.$$
where *S_i_* is the confidence score of the current prediction frame, *M* is the prediction frame with the highest confidence score, *b_i_* is the prediction frame in the current category, IoU represents the ratio of the intersection and union of two prediction frames, *N_t_* is a manually set threshold, generally 0.5; *σ* is the continuous penalty term coefficient. The above formula indicates that the prediction frame with the highest score is set as a reference frame by Soft NMS; then, the IoU (intersection over union) is calculated according to the reference frames and the remaining prediction frames in the current category, and predictive frames with IoU less than the set threshold are retained. When IoU is greater than the set threshold, the confidence score of the prediction frame will continuously reduce, instead of directly setting it to zero, some frames with high scores can be used as correct detection frames in subsequent calculations. Therefore, the ability of detecting occluded overlapping fruits can be effectively improved. At the same time, the Soft NMS Algorithm is not more complex than traditional NMS and is easy to implement.

### 4.4. Improvement of Network Model

The improved network structure is shown in Figure 5. The backbone network is composed of five CSPX modules with 1, 2, 8, 8, and 4 residual blocks, respectively, and each also contains two CBL modules. CBL module is a combination of BN layer, activation layer and convolution layer, the activation function still uses the Lekey Relu from the original network. The pixel of the input image is set to 608 × 608, after each CSPX module, the feature map is reduced to 50% of its original size. After 32 times down sampling through the backbone network, seven CBL modules and one SPP module in the neck network, as well as the final convolution layer, a prediction feature figure y1 with a size of 19 × 19 is obtained for predicting larger targets; then, after passing through 6 CBL modules and 1 SPP module, the output feature map is subjected to two times of up sampling operations in turn, and the results are spliced and fused with the shallow feature map with lower resolution according to the channel. Finally, two predicted feature maps y1 and y2 with different scales are obtained (their sizes are 38 × 38 and 76 × 76), which can predict medium-sized and small objects, respectively; each prediction feature map has 21 channels (each feature point on the feature map can predict three priori frames, each priori frame performs two category predictions, one confidence prediction and four position predictions; therefore, a total of 21-dimensional prediction feature vectors are required). Finally, the prediction results from the feature maps with three scales are sent to Soft NMS together, and the final prediction is completed after filtering out the redundant prediction frames.

### 4.5. Optimization of Loss Function

#### 4.5.1. Focal Loss [33]

The problem of sample imbalance exists in first-level networks such as YOLOv5. Negative samples may interfere with the optimization direction of the model during training. The weight of all samples is the same in the standard cross entropy loss function; therefore, if the samples are unbalanced, the negative samples will occupy the dominant position, and a small number of difficult to detect samples and positive samples will not work, which will lead to poor accuracy, as shown in Equation (2):(2)$$FL\left(p\right)=\left\{\begin{array}{c} -\alpha {\left(1-p\right)}^{\gamma }\log\left(p\right),y=1\\ -{\left(1-\alpha \right)p}^{\gamma }\log\left(1-p\right),else \end{array}\right.$$
where p represents the sample’s prediction probability, *y* represents the sample label, *α* is the balance coefficient of the samples, 0 < *α* < 1; *γ* is a modulation factor of the difficult and easy sample.

The *α* is introduced in Equation (2) to balance the weights of the samples, and (1 − *p*)*^γ^* can adjust the weights of difficult and easy samples. Taking the prediction of positive samples (*y* = 1) as an example, when a border is misclassified, if *p* is small, then (1 − *p*)*^γ^* is close to 1, and its loss is hardly affected, when *p* is close to 1, which shows that the classification prediction is better and it is a simple sample, (1 − *p*)*^γ^* is close to 0, the loss is reduced, and the contribution of simple sample loss will decrease with the increase in *γ*.

#### 4.5.2. CIoU Loss (Complete Cross-Over Loss)

The position loss in the original YOLOv5 loss function is obtained by directly calculating the square of the deviation between the four prediction offsets and the actual offsets; it does not take into account that the overlap area and overlap position between the real box and the predicted box may be different under the same offset loss value. Therefore, we propose to use CIoU Loss (complete IoU Loss) to replace the L2 position loss in the original loss function. Firstly, the prediction frame can be obtained by inputting the prediction offset into the priori frame. Then, we directly minimize the distance between the center points of the prediction box and the real box, and maximize the overlapping area between the two boxes, as shown below.
L_CIoU_ = 1 − CIoU(3)
where
$$\mathrm{CIoU}=\mathrm{IoU}-\left(\frac{d^{2}}{c^{2}}+uv\right)$$
$$v=\frac{4}{\pi^{2}}{\left(\arctan\frac{w^{gt}}{h^{gt}}-\arctan\frac{w}{h}\right)}^{2}$$
$$u=\frac{v}{(1-\mathrm{IoU})+v}$$

In the above equation, *d* represents the distance between the center points of the prediction box and the real box, *c* represents the diagonal distance of the minimum bounding rectangle of the prediction box and the real box, *w^gt^* is the width of the real box and *h^gt^* is the its height.

According to Formula (3), when the predicted box is completely coincident with the real box, IoU = 1, *d* = 0, *uν* = 0, So CIoU = 1, L_CIoU_ = 0, and when the distance between the prediction box and the real box is infinitely far, IoU = 0, *d*^2^/*c*^2^ = 1, CIoU = −(1 + *uν*), so L_CIoU_ = 2 + *uν*, 0 < L_CIoU_ < 2 + *uν*, where v is used to indicate the similarity of the aspect ratio and u is the weight coefficient. Since CIoU Loss comprehensively considers the above factors, the regression of the target frame is more stable, the convergence rate is faster, and the position prediction is more accurate. Even if the prediction box does not intersect with the real box, the back propagation can be completed to optimize the model.

## 5. Model Training

### 5.1. Experimental Conditions

Tests were carried out in a deep learning environment on a workstation, configured as follows: Intel Xeon(R) E5-2650 v4@2.20 Hz × 48 CPU; Running Memory 64 GB; 1 TB Solid State Drive; 12 GB GTX1080Ti × 2 GPU; Ubuntu 18.04 operating system; NVIDIA drives 450.102.04; Python and Pytorch are 3.8 and 1.7, respectively; CUDA and cuDNN are 11.0 and 7.6.5, respectively.

### 5.2. Evaluation Index System

Model performance was evaluated via P, R, F1, AP and MAP, as shown in Equations (4)–(8).
(4)$$P=\frac{TP}{TP+FP}$$
(5)$$R=\frac{TP}{TP+FN}$$
(6)$$F1=\frac{2PR}{P+R}$$
(7)$$AR=\int_{0}^{1}P(R)dR$$
(8)$$MAP=\frac{\sum_{n=1}^{2}AP(n)}{2}$$

In Formulas (4)–(8), P is the ratio of correct prediction boxes to all prediction boxes, R is the proportion of correct label frames in all label frames, TP is the number of prediction boxes correctly matched by labeled boxes, FP represents the number of prediction boxes with incorrect prediction results, FN indicates the number of label frames that have not completed the detection task, and AP represents the average accuracy value for each apple class; MAP is the average accuracy value for both apple classes; F1 represents the harmonic mean of P and R.

The correct bounding box is a rectangle that accurately covers the outline of the apple (coverage is 99% or greater).

### 5.3. Training Process

We use SGD optimization algorithm with momentum; before the training, the improved model was pre-trained using a COCO dataset, the model was initialized using a pre-trained weight [34], and the Batch scale was set to 16. During training, we use a small initial value of learning rate at the beginning of training to avoid over fitting. In this article, the initial value is 0.001; after three rounds of training, it rose to 0.01, and gradually declined to 0.001 from round 4; the epoch is 300, and the weight decay is 0.0005. After each epoch is trained, the network weight file is saved, and the monitoring of the model is carried out in real time by using the Tensor board; each performance metric under the validation set is recorded. The training lasted for six hours and thirty-two minutes.

### 5.4. Analysis of Training Data

The loss curve during the iterative training and the performance index curve under the verification set are shown in Figure 6. Although the number of training rounds set in this experiment is 300 rounds, the loss curve tends to be stable after about 100 rounds, which indicates that the model has been fitted at this time. After 150 epochs, the loss curve fluctuates up, and training losses are still falling; this indicates an over fitting of the model. The curves of P, R, F1 and MAP in the training process show that the performance indicators change greatly but the overall trend is rising in the first 20 rounds of training; as the training progresses, it gradually becomes stable and oscillates in a small range. Since the model has been fitted in about 100 rounds, and the precision and recall rates of the two categories are comprehensively considered in the MAP value, this paper takes the maximum MAP near round 100; the maximum value of MAP was 96.1%, which occurred in round 109. That is, the weights obtained in this round are taken as the final weights of the model, where P = 94.1%, R = 90.6%; F1 = 92.3%.

## 6. Test Results

### 6.1. Evaluation of Test Results

The resulting model is tested under the test set. The test set contains two types of apple images in various complex environments. The P-R curve in Figure 7a shown that the AP values of ripe apples and unripe apples reached 96.4% and 96.3%, respectively; the MAP is 96.3%, the average detection speed is 27.8 frames/s, and the performance index meets requirements of the picking robot. F1 represents the harmonic mean of accuracy and recall, which changes with the change in the confidence threshold, F1 curve is shown in Figure 7b. When the confidence threshold is 0.674, the maximum value of F1 is 0.92. Since a large threshold will lead to a decrease in recall rate and an increase in missed detection rate, the confidence threshold is set to 0.55 after comprehensive consideration of accuracy rate and recall rate. In this case, the model has the best performance.

### 6.2. Recognition Results of Different Algorithms

The other five target recognition algorithms are trained under the dataset, and after the optimal model was obtained, tests are performed on test sets. The average value of the accuracy, recall rate, F1, AP and detection speeds were shown in Table 1. Among them, the performance of the improved recognition algorithm is as follows: accuracy P = 91%, recall rate R = 92.6%, F1 value = 91.8%, AP1 = 96.4%, AP2 = 96.3%; all the performance indexes are better than those of the other five recognition algorithms. The AP1 and AP2 of improved YOLOv5 were increased by 2.2% and 5.5%, respectively, and 1.3% higher than those of YOLOv5. The above test results show that the improved algorithm has better comprehensive recognition performance.

The recognition results in Figure 8 show that the improved algorithm can well detect mature and immature apples in various complex environments such as different illumination, different occlusion degrees, bagging, and large field of view. The position and category of the prediction box are more accurate, the rate of missed and false detection is lower, and the robustness is stronger.

### 6.3. Comparative Experiment under Different Fruit Number

The following is a comparative test under different number of apples. The test set has a total of 400 images, including 5117 apples, which are divided into four categories according to the number of apples. Of these, 105 were images of a single apple, 142 images with multiple apples (each with 2 to 10 apples), 125 images of dense apples (each containing 11 to 30 apples), and 28 images of apples in a large field of view (each contains more than 30 apples). The performance of the two methods under different apple numbers is shown in Figure 9; this result shows that the two algorithms have better recognition results when detecting a single apple and multiple apples. When detecting dense apples, the original YOLOv5 algorithm can cause missed detection and inaccurate positioning. In a complex environment, the number of missed and falsely detected apples in YOLOv5 is significantly more than that of the improved algorithm.

The performance indexes obtained from the above tests are shown in Table 2. When detecting a single apple, there is little difference between the improved algorithm and YOLOv5. With the increase in fruit quantity, the performance indexes of the two algorithms decreased. Moreover, the gap in recognition accuracy between YOLOv5 and the algorithm in this paper is becoming more and more obvious. Among them, when detecting multiple apples, compared with the improved algorithm, the AP values of the two classes are reduced by 1.5% and 3.5%, respectively, and the F1 value is 2.0% lower. When detecting dense apples, the AP values of the two types were 1.8% and 4.0% lower than that of the proposed algorithm, and the F1 values were 3.7% lower. In the large field of view environment, the two types of AP values are 2.8% and 6.7% lower than the algorithm in this paper, and the F1 is 5.7% lower.

In summary, the improved algorithm can well identify apples under different quantity, size, overlap and occlusion environment, and its performance is significantly better than the original YOLOv5, especially for apple recognition in dense and large field of view environments, where the performance improvement is even more significant; the effectiveness of the improved algorithm has been fully verified.

### 6.4. Comparison Test under Different Light Conditions

In this section, the robustness of the improved algorithm is tested under different lighting conditions, which include front light, side light, backlight (including evening and cloudy days, etc.) and artificial lighting at night, as shown in Figure 10, the results show that the two types of algorithms have better recognition results under different lighting conditions; however, when the light is extremely poor (such as backlight, evening, etc.), the prediction frame position is not accurate enough and false detection and missed detection will occur. The performance indexes obtained from the above tests are shown in Table 3. The above results show that the performance indexes of the two algorithms have little difference under the conditions of front light, side light, backlight and artificial light at night, and good detection results can be achieved, which indicates that this recognition algorithm has strong robustness to different illumination conditions.

## 7. Conclusions

In the complex environment of orchards, in view of low fruit recognition accuracy, poor real-time and robustness of traditional recognition algorithms, this paper proposes an improved fruit recognition algorithm based on deep learning. The P of the algorithm in the test set is 91%, R is 92.6%, F1 is 91.8%, the AP of the two categories are 96.4% and 96.3%, respectively, and the recognition speed can reach 27.8 frames/s. The overall performance is better than the five currently known mainstream algorithms and can meet the requirements of automatic apple picking in accuracy and real time.

To further verify the effectiveness and feasibility of the improved algorithm, the performance comparison test of the algorithms was carried out under different fruit quantities and light conditions; the test results show that the two types of AP values are improved by 1.8% and 4.0%, respectively, when recognizing dense apples, and they are improved by 2.8% and 6.7%, respectively, in the large field of view environment, which proves the superiority of the improved algorithm. In addition, this target recognition algorithm has strong robustness to lighting conditions, and the change in light has little influence on its recognition performance.

## Figures and Tables

**Figure 1 sensors-23-05425-f001:**
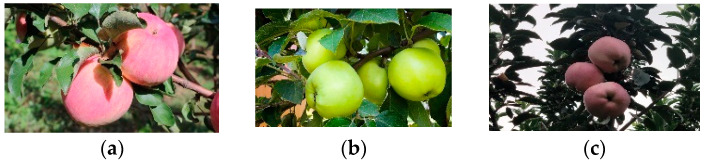
Apple images in complex scenes. (**a**) Natural light; (**b**) Side light; (**c**) Back light; (**d**) Immature bagging; (**e**) Mature overlapping occlusion; (**f**) Immature overlapping occlusion; (**g**) Mature bagging; (**h**) Artificial lighting at night; (**i**) Wide field.

**Figure 2 sensors-23-05425-f002:**
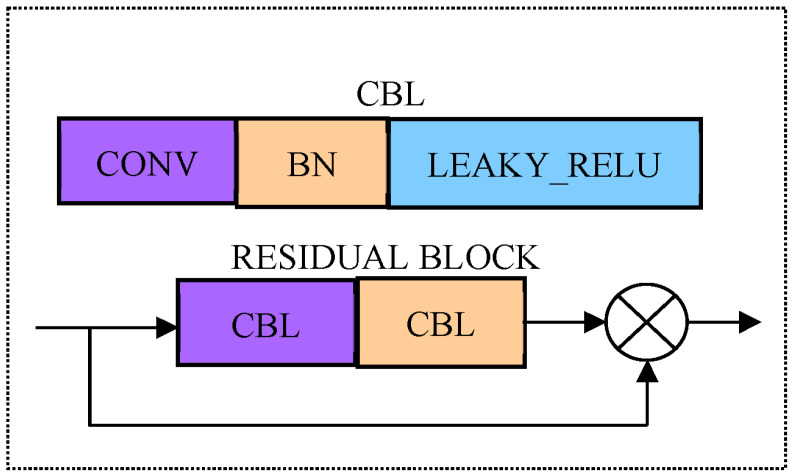
Residual block of YOLOv5 model. Note: Conv represents convolution; CBL represents the synthesis module of Conv add Batch Normalization add Lekey ReLU activation function; Each residual block contains two CBL modules, where 1 × 1 and 3 × 3 is the size of convolution kernel in Conv; Add is the addition of tensors and does not expand the dimension.

**Figure 3 sensors-23-05425-f003:**
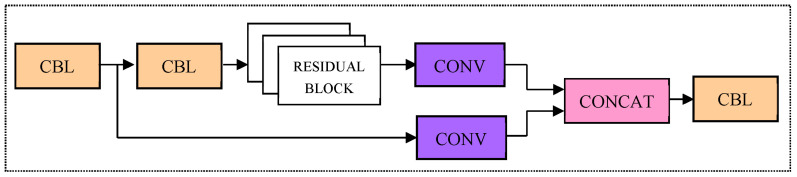
CSP_X module. Note: CSPX represents a CSP module with *X* residual block inside, the convolution kernel in the first CBL of each CSP is 3 × 3, and stride = 2, so it can perform a down-sampling operation.

**Figure 4 sensors-23-05425-f004:**
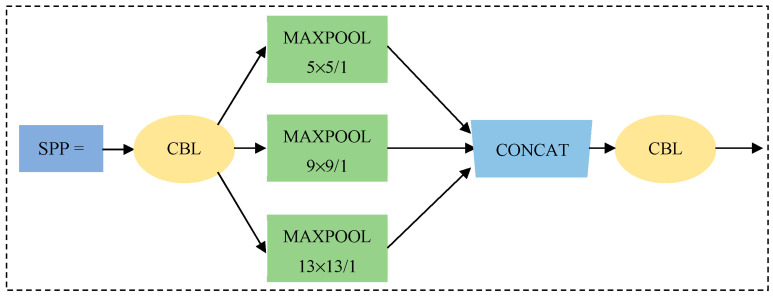
Spatial pyramid pooling (SPP) module. Note: Maxpool represents the maximum pool, where 5 × 5, 9 × 9 and 13 × 13 represent the size of the pool kernel of the pool layer, /1 represents the stride = 1.

**Figure 5 sensors-23-05425-f005:**
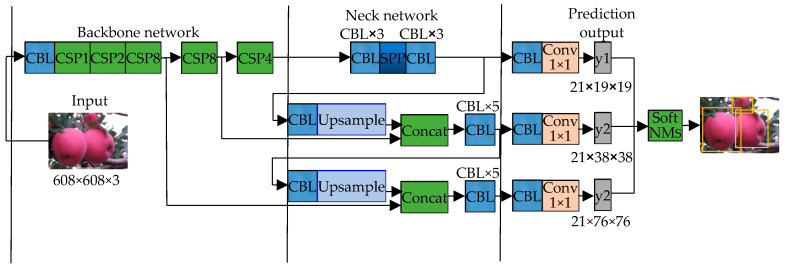
Improved network structure diagram. Note: The number after CSP indicates that the CSPX module contains several residual blocks; Concat is a feature fusion method based on superposition of channel numbers; up sample means up sampling the input feature map; y1, y2 and y3 represent output characteristic diagrams of three different scales, respectively; The ×*i* representation in the diagram should have *i* multiple of same module composition here.

**Figure 6 sensors-23-05425-f006:**
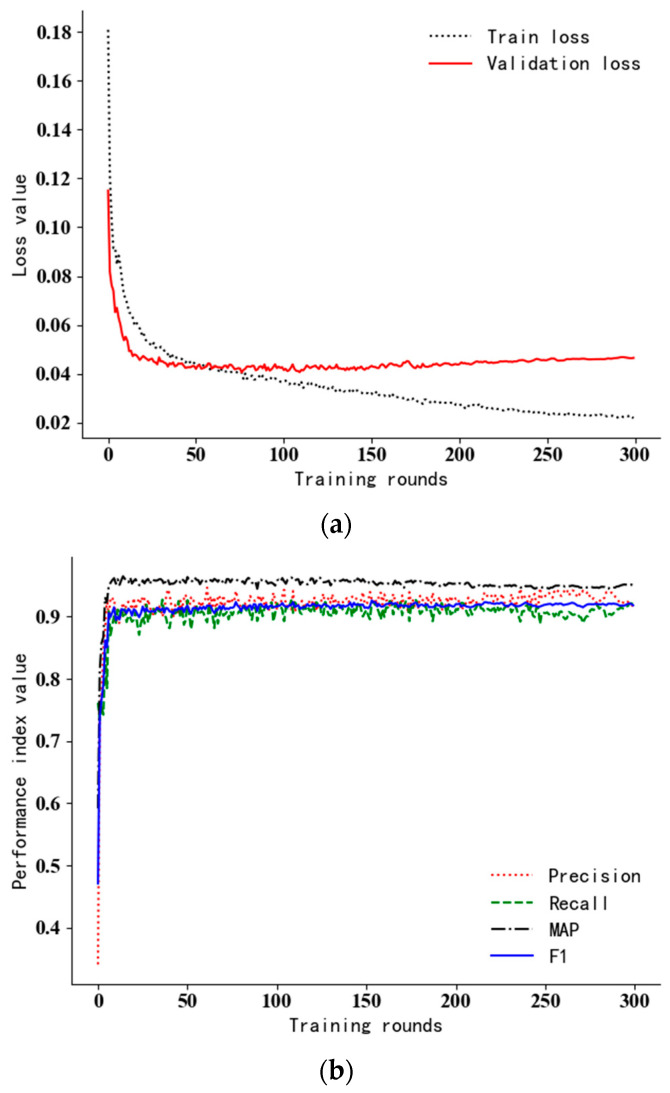
The loss curve and the indexes of the verification set in the training process. (**a**) The loss curve; (**b**) various performance indexes.

**Figure 7 sensors-23-05425-f007:**
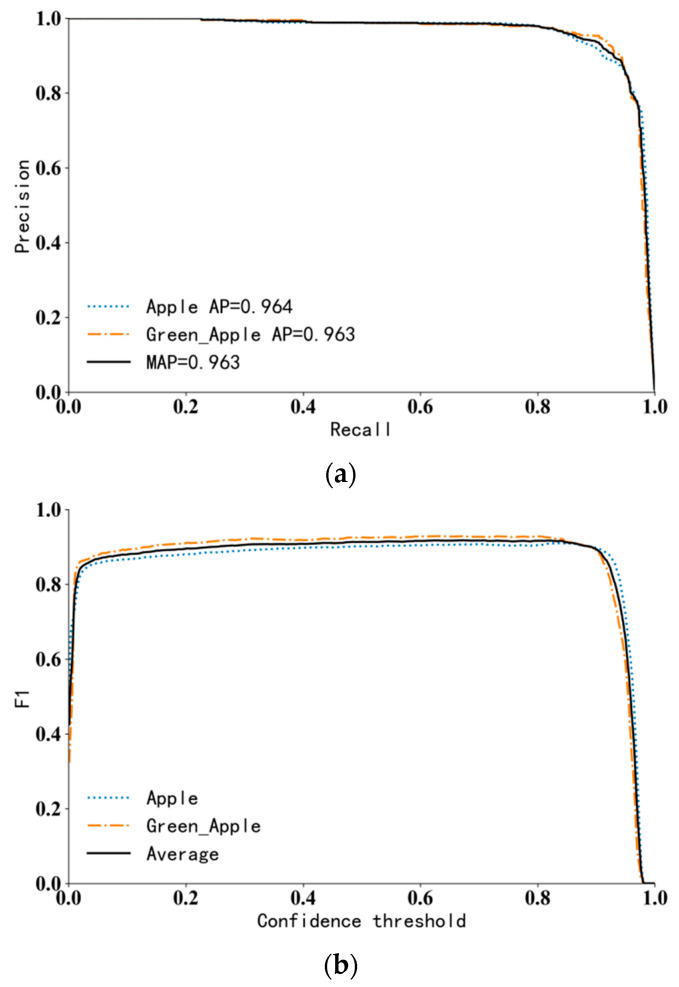
P-R and F1 curve under test set. (**a**). P-R curve; (**b**). F1 curve. Note: apple represents the category of mature apples, green apple represents the category of immature apples, AP represents the average precision value of a single category, and MAP represents the average precision value of two categories.

**Figure 8 sensors-23-05425-f008:**
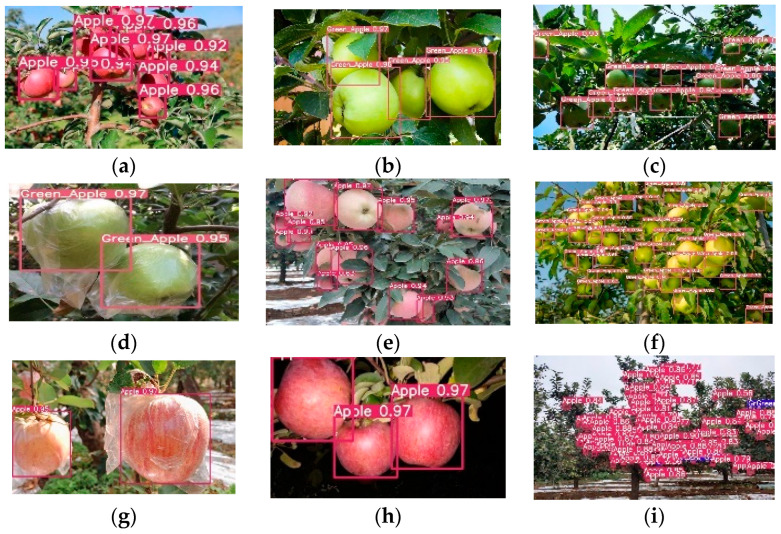
Detection effect of this algorithm. (**a**). Natural light; (**b**) side light; (**c**) back light; (**d**) immature bagging; (**e**) mature overlapping occlusion; (**f**) immature overlapping occlusion; (**g**) mature bagging; (**h**) artificial lighting at night; (**i**) wide field.

**Figure 9 sensors-23-05425-f009:**
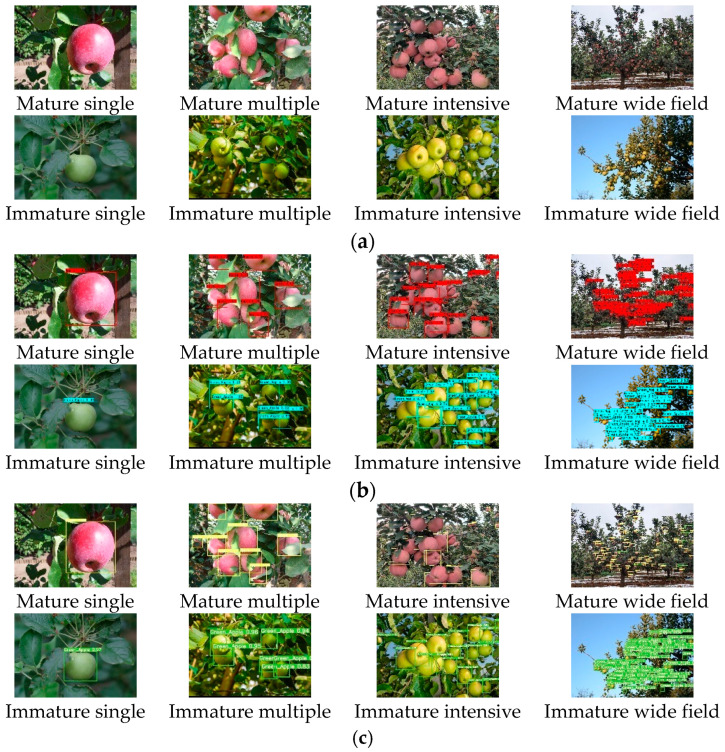
Detection effect of different numbers of apples before and after YOLOv5 improvement. (**a**) Original picture; (**b**) YOLOv3; (**c**) improved YOLOv5.

**Figure 10 sensors-23-05425-f010:**
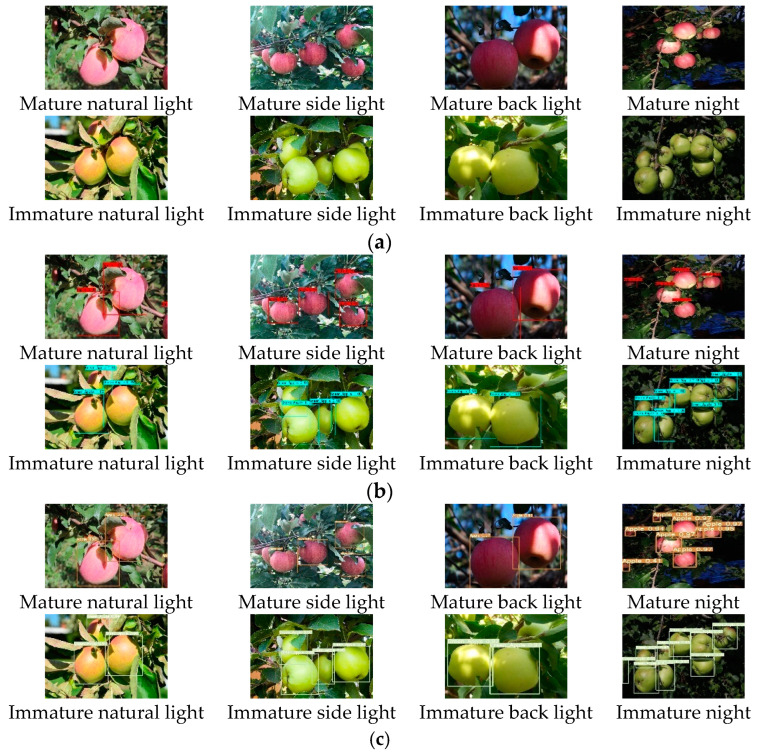
Detection effect of apples under different illumination before and after YOLOv5 improvement. (**a**). Original picture; (**b**). YOLOv5; (**c**). Improved YOLOv5.

**Table 1 sensors-23-05425-t001:** Recognition results of six algorithms on the test set.

Detection Algorithm	P/%	R/%	F1/%	AP1/ %	AP2/ %	Detection Speed F·s^−1^
Faster RCNN	91.5	73.1	81.0	92.5	85.2	16.5
RetinaNet	89.8	81.8	85.5	92.7	88.6	26.3
CenterNet	90.4	70.3	79.0	90.7	83.6	32.3
YOLOv5	91.7	92.0	91.8	95.1	95.0	25.6
Improved YOLOv5	91.0	92.6	91.8	96.4	96.3	27.8

Note: AP1 is the AP value of mature apple, and AP2 is the AP value of immature apple.

**Table 2 sensors-23-05425-t002:** Recognition results of different numbers of apples before and after YOLOv5 improvement.

Apples Number	Algorithm	P/ %	R/ %	F1/ %	AP1/ %	AP2/ %
Single	YOLOv5	92.4	98.9	95.5	98.8	99.6
	Improved YOLOv5	98.1	95.7	96.9	98.5	98.8
Multiple	YOLOv5	89.6	91.2	90.5	95.3	93.4
	Improved YOLOv5	91.9	93.1	92.5	96.8	96.9
Intensive	YOLOv5	89.9	85.2	87.5	94.2	91.0
	Improved YOLOv5	89.6	92.8	91.2	96.0	95.0
Wide field	YOLOv5	84.4	82.5	83.0	90.4	85.6
	Improved YOLOv5	85.4	86.6	88.7	93.2	92.3

**Table 3 sensors-23-05425-t003:** Recognition results of apples under different illumination.

Light Condition	Detection Algorithm	P/ %	R/ %	F1/ %	AP1/ %	AP2/ %
Natural light	YOLOv5	91.3	95.4	93.3	97.3	96.8
	Improved YOLOv5	95.2	94.7	94.9	97.7	97.7
Side light	YOLOv5	91.6	95.6	93.6	97.4	97.0
	Improved YOLOv5	95.6	94.8	95.2	98.0	97.9
Back light	YOLOv5	90.7	94.5	92.6	96.9	96.0
	Improved YOLOv5	94.6	94.0	94.3	97.3	97.2
Night	YOLOv5	90.8	95.0	92.9	97.0	96.5
	Improved YOLOv5	95.0	94.2	94.6	97.4	97.6

## Data Availability

The data that support the findings of this study are included within the article.

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
