# Peer review of "Research on Apple Recognition Algorithm in Complex Orchard Environment Based on Deep Learning"

_sensors, 2023, doi:10.3390/s23125425_

Round 1

Reviewer 1 Report

In this paper authors present a neural network for fast and accurate detection of apples (2 class detection – for mature and immature fruits) in a complex natural orchard environment. This solution may have significant importance for robotic systems, working in orchards.  

  • The results in the Abstract are too general: “recognition accuracy was significantly improved”, and “algorithm is excellent in recognition accuracy, robustness and real-time performance”. Please provide specific numbers on known benchmarks.

  • In the introduction: “Therefore, the research on algorithms for accurately identifying fruits on trees in complex environments has important academic significance and practical application value.” The above sentences highlight only practical value, not academic significance. Please, elaborate on the academic part.

  • The major part of your introduction is in fact a description of related literature, in which you mention different object detectors and their applications to the task of fruit detection. Since there is no related work section in your paper, you should also overview network- and detention-improving techniques that you use (CSP, SPP, soft NMS), and probably some others in the Introduction section.

  • Please, clearly state your contributions at the end of the Introduction section.

  • You should also provide an outline of the paper at the end of the introduction.

  • “...some apple images of different varieties … are collected through the Internet”. There might be some problems with copyright, please ensure that there are none and state so in your paper.

  • In section 2.1 called “Original image processing” you in fact describe the data collection process and image sources, not the processing. Please correct the title.

  • In section 2.3 you describe mosaic data enhancement, which is very similar to YOLOv4 (see Bochkovskiy, Alexey, Chien-Yao Wang, and Hong-Yuan Mark Liao. "Yolov4: Optimal speed and accuracy of object detection." ). Proper citation is required. If your method is somehow different from the original, please describe how.

  • “...the maximum pooling layer is replaced by a 3×3 convolution with step size 2 for downsampling.” This “step size” is usually called “stride”.

  • There is no reference to Figure 2 in the text.

  • “Cspnet (Cross stage parallel network) is an advanced optimization method which can eliminate gradient information repetition.” The poor word choice here. CSPNet is not an optimization method (like SGD, Adam, etc) it is a neural network. What you want to describe here is Cross-Stage-Partial-connections – basic blocks of CSPNet.

  • Both CSP and SSP were previously used in object detection as parts of the YOLOv4 network (see the original paper, mentioned above). Please refer to it and, if there are differences in your implementations – clearly state them.

  • The soft NMS algorithm is described, but the original paper (Bodla, Navaneeth, et al. "Soft-NMS--improving object detection with one line of code”) is not cited. Please, cite it.

  • In section 3.5.1 Focal Loss. Once again you don’t cite the original paper (“Lin, Tsung-Yi, et al. "Focal loss for dense object detection.”). You should.

  • In the formula (4)-(6) (Precision, Recall, F1-score). You write that you compute those scores box-wise. But what is a correct bounding box? Is it a direct match with a corresponding ground truth or do you allow for small shifts? And is so – for how small? Please describe it in section 4.2.

  • In section 4.3 Training process. Please, identify which optimization algorithm you use (SGD, SGD with momentum, Adam, or any other) and with which parameters.

  • Table1. Please, use bold markup for the best results in each column. And provide citations for all the other models, which you compare to.

  • Since you aim for a robotic solution, you should measure the detection speed of your network (and the other networks in Table 1) using the appropriate device. Most likely – low-power CPU, because it is doubtful that a robot would have GTX1080Ti GPU onboard.

  • It is hard to see small bounding boxes in Figures 8 and 9 at a given resolution.

Since the detection speed is essential and the network will be used on a robot you should consider network quantization (replacement of floating-point weights with integers). Usually 8-bit quantization does not introduce any noticeable accuracy gap while significantly boosting the performance and reducing the memory footprint.

Finally, professional English editing might be a good idea.

professional English editing might be a good idea.

Reviewer 2 Report

It is important to clearly discuss the scientific area to which a contribution is required in the writing, and it is necessary to clearly write the methods and algorithms that will be used in the research, as this will help readers understand the results. The abstract should have a logical sequence that allows us to know the general background, the scientific area that will be discussed, the knowledge gap considered as a bias to correct or improve.

Finally, in the results section, from line 431 to 438, we find fragments that can be written in the abstract to strengthen it.

To achieve an effective and accurate identification of different degrees of maturity of the apple, it would be helpful to design a global diagram of the methods used for specific actions, since as we know, in a written report, not all activities are regularly reported due to space constraints.

Best regards

Check the grammar: make sure that sentences are well-constructed and that verbs are in the correct tense.
Simplify the language: try to use clear and simple language that is easy to understand for readers. Avoid using complex or technical words or phrases that may confuse the reader.
The ideas in the article should be organized into logical sections and use subheadings to help the reader follow the structure of the article.
Ensure that each section is well-defined and that the transition from one section to another is clear.

Reviewer 3 Report

Aiming at the rapid and accurate identification of apple fruits of different maturity under 11 different lighting, overlapping occlusion and large field of view, this paper proposes an innovative 12 apple fruit recognition algorithm by improving YOLOv5. This paper has certain novelty, but some problems should be addressed:

1.     The reviewed works are very old. Also, the compared methods are old. YOLOv8 has been proposed. Why the authors do not compare with YOLOv8?

2.     Many works about deep learning methods should be mentioned, e.g,

Motion Stimulation for Compositional Action Recognition;

Recurrent Thrifty Attention Network for Remote Sensing Scene Recognition;

Real-time multi-class helmet violation detection using few-shot data sampling technique and yolov8;

Deep learning in optical metrology: a review;

Deep learning, reinforcement learning, and world models;

Video moment retrieval with noisy labels

……

3.     The details on the datasets used in the experiments are not clear. For example, how to take the photos? The distance? The season? ……

4.     Lack of more explanations on why your proposed steps are so-suitable for the apple recognition. The authors should add more explanations when presenting new models.

5.     Please put figures and their captions in a page.

6.     I am very concerned with the problem that how to label the data.

Round 2

Reviewer 3 Report

Suggest accepting this paper for publishing in this journal.

NO